# Hierarchical Agents by Combining Language Generation and Semantic Goal Directed RL

**Bharat Prakash**[1], **Nicholas Waytowich**[2], **Tim Oates**[1], **Tinoosh Mohsenin**[1]
University of Maryland, Baltimore County[1]
US Army Research Lab [2]

## Abstract

Learning to solve long horizon temporally extended tasks with reinforcement learning has been a challenge for several years now. We believe that it is important to leverage both the hierarchical structure of complex tasks and to use expert supervision whenever possible to solve such tasks. This work introduces an interpretable hierarchical agent framework by combining sub-goal generation using language and semantic goal directed reinforcement learning. We assume access to certain spatial and haptic predicates and construct a simple and powerful semantic goal space. These semantic goal representations act as an intermediate representation between language and raw states. We evaluate our framework on a robotic block manipulation task and show that it performs better than other methods, including both sparse and dense reward functions. We also suggest some next steps and discuss how this framework makes interaction and collaboration with humans possible.

## 1   Introduction

Deep reinforcement learning has been successful in many tasks, including robotic control, games, energy management, etc. Mnih et al. [2015] Schulman et al. [2017] Warnell et al. [2018]. However, it has many challenges, such as exploration under sparse rewards, generalization, safety, etc. This makes it difficult to learn good policies in a sample efficient way. Popular ways to tackle these problems include using expert feedback Christiano et al. [2017] Warnell et al. [2018] and leveraging the hierarchical structure of complex tasks. There is a long list of prior work which learns hierarchical policies to break down tasks into smaller sub-tasks Sutton et al. [1999] Fruit and Lazaric [2017] Bacon et al. [2017]. Some of them discover options or sub-tasks in an unsupervised way. On the other hand, using some form of supervision, either by providing details about the sub-tasks, intermediate rewards or high-level guidance is a recent approach Prakash et al. [2021] Jiang et al. [2019] Le et al. [2018].

This paper presents a framework for solving long-horizon temporally extended tasks with a hierarchical agent framework using semantic goal representations and goal generation using language. The agent has two levels of control and the ability to easily incorporate expert supervision and intervention. The high-level policy is a small text generation model which generates sub-goals in the form of text commands, given a high level goal and current state. The low-level policy is a goal-conditioned multi-task policy which is able to achieve sub-goals where these goals are specified using a semantic goal representation. There is an intermediate module which converts these text goals to semantic goal representation. The semantic goal representation is constructed using several predicate functions which define the behavior space of the agent. This representation has many benefits because it is much simpler than traditional state-based goal spaces as shown in Akakzia et al. [2020]. The language interface makes the framework more interpretable and easier for an expert to intervene and provide high-level feedback. The sub-goals which are in the form of language can be observed by a human expert and they may provide corrections if necessary.

36th Conference on Neural Information Processing Systems (NeurIPS 2022).

We evaluate the framework using a robotic block manipulation environment. Our experiments show that this approach is able to solve different tasks by combining grasping, pushing and stacking blocks. Our contributions can be summarised as follows:

- A hierarchical agent framework where the high-level policy is a language generator and the low-level policy is learned using semantic goal representations.
- A language interface that can map natural language commands to symbolic goals. This module is also a natural interface humans to intervene and provide corrections.
- Evaluation on complex long horizon robotic block manipulation tasks to show feasibility and sample efficiency

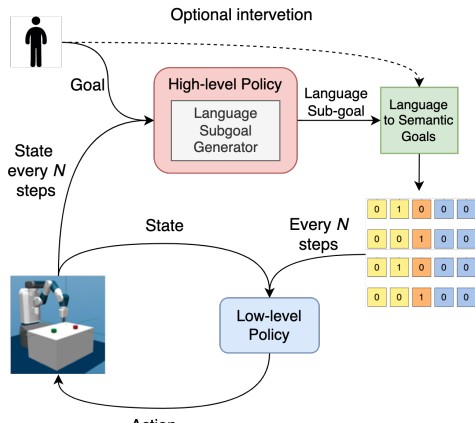

Figure 1: The low-level policy receives semantic goals and is trained to execute primitive actions in the environment to achieve the goal. Given a high level goal, the high-level policy outputs a sub-goal in terms of language commands. This is then converted into semanntic goals before feeding it to the low-level policy.

## 2 Methods

In this section, we present a framework for solving long horizon temporally extended tasks. We first describe the semantic goal representation and low-level policy training. Then, we show how the high-level policy is obtained using the sub-goal instruction generator and solve long horizon tasks.

### 2.1 Semantic goal representations

We represent goals using a list of semantic predicates which are determined based on domain knowledge. In our case we consider three spatial predicates - *close*, *above*, *in-bin* and one haptic predicate - *holding*. As demonstrated by Akakzia et al. [2020], these predicates define a much simpler behavior space instead of the traditional more complicated state space. This representation eliminates the need to write reward functions for every desired behavior. All these predicates are binary functions applied to pairs of objects. The *close* predicate is order-invariant. $close(o_1, o_2)$ denotes whether objects (in our case blocks) $o_1$ and $o_2$ are close to each other or not. The *above* predicate is applied to all permutations of objects. $above(o_1, o_2)$ is use to denote if $o_1$ is above $o_2$. The *in-bin* predicate is used to denote whether the block is inside the bin. Finally, *holding* is used to denote if the robot arm is holding an object using $holding(o)$. With these predicates we can form a semantic representation of the state by simply concatenating all the predicate outputs as shown in Fig 1.

### 2.2 Training the low-level policy

The low-level policy is trained to perform several individual sub-tasks, which can eventually by used to solve longer high-level tasks. We use Hindsight experience replay (HER) Andrychowicz et al. [2017] along with Soft-Actor critic (SAC) Haarnoja et al. [2018] to train the goal conditioned policy. Goals are sampled from a set of configurations based on the environment where an expert can be used to optionally create a curriculum. The semantic goal representation makes is easier to do both of these things. The agent explores the environment to collect experience and updates its policy using SAC. As stated earlier, there is no need to write reward functions for each desired behavior. A reward can be generated by checking whether the current semantic configuration matches the goal configuration. Example sub-goals for the three environments we use are listed in Table 1.

### 2.3 Training the high-level policy

The high-level policy is a sub-goal instruction generator which takes in the current state and high-level task description and outputs a sub-goal in the form of a language instruction. It is trained using a

| 2 Blocks | 3 Blocks | Desk Cleanup |
|---|---|---|
| pick up the red block | pick up the blue block | grasp red block |
| grab green block | put blue block close to red block | drop red block in the bin |
| put red block close to green block | drop green block away from blue block | put green block on the table |
| stack red block above green block | | |

Table 1: This table shows the sub-goals used in our tasks. The semantic goal representation is built using these sub-goals as described in the previous section. The high level policy generates subgoals from the above set of phrases as described in section 2.3

small dataset of demonstrations where the sub-tasks are labeled using language instructions by human experts. The sub-goal instruction generator is a neural network which receives the current state, $s_i$ and the high level goal $L_{hg}$ and is processed using a recurrent neural network. It then uses another recurrent neural network to output a sub goal $L_{sg}$. The dataset consists of high-level goal, current state and sub-goal tuples. We use around 300-500 ( 100 trajectories) samples each of the tasks. The language phrases are replaced with synonyms and similar phrases. The dataset size increases $5\times$ after this simple data augmentation. The language goal is then converted into semantic goals, which is used to train the low-level policy using a small module which its trained using the same dataset. The low-level policy performs N environment steps before the high-level takes control and provides a new sub-goal.

## 3 Experiments

### 3.1 Environment setup and tasks

We design two versions of the Fetch manipulation environment with 2 and 3 blocks.
**2 blocks environment** Here we have the robotic arm as mentioned earlier and two blocks: red and green. We consider all three predicates for this version, *close*, *above* and *holding*. We design 3 high-level tasks in this environment (1) Move blocks close: Here the task is initialized with the 2 blocks far away from each other. The goal is to bring them close to each other. (2) Move blocks apart: Here the task is initialized with blocks close to each other. The goal is to move them apart. (3) Swap blocks: Here the task is initialized with blocks on top of each other in random order. The goal is to swap the order.
**3 blocks environment** Here we have the robotic arm and 3 blocks: red and green and blue. For this version, we only consider 2 predicates, *close* and *holding*. We design 2 high-level tasks in this environment (1) Move blocks close: Here the task is initialized with all the 3 blocks far away from each other. The goal is to bring them close to each other. (2) Move block apart: Here the task is initialized with blocks close to each other. The goal is to move them apart.
**Desk cleanup environment** Here we have a robotic arm several blocks on the desk. The desk also a bin and the blocks are places randomly on the desk. The task is to clean up the desk and place all the blocks inside the bin. We use 2 predicates here, *holding* and *in-bin*. We have 3 versions with 2, 3 and 4 blocks.

### 3.2 Baselines

*1. Flat semantic*: Here the agent has a single level flat policy but the goals are still represented using the semantic goal representations. *2. Flat Continuous*: Here the goals are represented using the actual block positions of the desired configuration. The dense reward function is based on the distance between current and desired block locations and hence it is a dense reward function. *3. Option Critic*: This is a hierarchical reinforcement learning baseline. It has two levels of control where we can provide the number of desired options. *3. H-Planner*: This is a hierarchical planning agent where the architecture is similar to our but the high level policy is a STRIPS planner Alkhazraji et al. [2020] which outputs a high-level plan. This is then executed sequentially by the low-level policy.

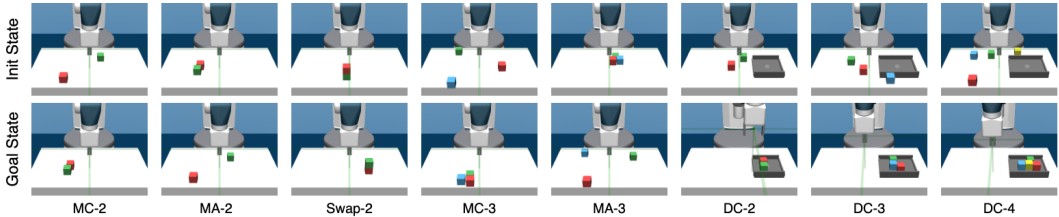

Figure 2: This figure show all the tasks we used in our experiments. The top row shows examples of random initial states and bottom row shows the goal states. MC-2: Move 2 blocks closer, MA-2: Move 2 blocks away, Swap-2: Swap 2 stacked blocks, MC-3: Move 3 blocks closer, MA-3: Move 3 blocks away, DC-2 to DC-4: Desk clean up with 2, 3 and 4 blocks

| | | | | Tasks | | | | |
|---|---|---|---|---|---|---|---|---|
| **Method** | **MC-2** | **MA-2** | **Swap-2** | **MC-3** | **MA-3** | **DC-2** | **DC-3** | **DC-4** |
| Flat Semantic | 10% | 80% | 0% | 5% | 10% | 30% | 0% | 0% |
| Flat Continuous | 5% | 10% | 0% | 0% | 0% | 0% | 0% | 0% |
| Option Critic | 5% | 5% | 0% | 0% | 0% | 0% | 0% | 0% |
| H-Planner | **95%** | **100%** | **92%** | **95%** | **96%** | **94%** | **91%** | **90%** |
| **H-Lang (Ours)** | **91%** | **94%** | **90%** | **90%** | **90%** | **92%** | **90%** | **85%** |

Table 2: **Task completion %** This table shows the task completion % for our experiments. The tasks names are explained in Figure 4. As seen, our method consistently outperforms all the other baselines. We train each agent for 2M steps and roll out 50 episodes using the trained policy. The values are an average of runs from three different seeds.

## 3.3 Results

We calculate task completion % for all the tasks using the fully trained agent. We train each agent for 2M steps and roll out 50 episodes using the trained policy. The values are an average of runs from three different seeds. As seen in Table 2, only the 2 methods, H-Planner and our method is able to solve all the tasks. H-Planner uses an off the shelf planner which is makes use of the semantic predicates and produces a plan. the subtasks in the place are then sequentially executed by the low-level policy. Although it slightly outperforms our method, it is less interpretable and does not have a natural language interface for humans to intervene. Our method, denoted by H-Lang on the other hand, shows comparable performance and outperforms the other baselines.

## 4 Conclusion

In this paper we show that combining a high-level language generator, semantic goal representations and a low-level goal conditioned reinforcement learning policy is indeed a promising approach to build interpretable hierarchical agents. This also makes it easier for a human to intervene at the high-level to provided appropriate sub-goals using language in case there is a failure in the high-level policy.

There are several directions in which this framework can be extended. With the current state space, we assumed access to predicate functions. But with more complex observation like images, one can learn these predicate functions using a small amount of labelled data. To further demonstrate the capabilities of the framework we plan to perform experiments on more complex environments, real robots and qualitative analysis using human users. To measure the benefits of the language interface more systematically, user studies could be performed with humans and real robots to perform a qualitative analysis. This work is a step towards simple and interpretable hierarchical agents and we hope to build upon it.

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
