# OpenReview forum: "Hierarchical Agents by Combining Language Generation and Semantic Goal Directed RL"
_NeurIPS.cc/2022/Workshop/LaReL — LaReL 2022_

### Official Review · Reviewer_VpG6 · 2022-10-15
**Review of HLang**

**Rating:** 7
**Confidence:** 4

**Review:**

This paper presents a method for hierarchical goal-directed reinforcement learning. The method decomposes goals into language-based subgoals, such that the RL agent can solve long-horizon tasks. This concept is very similar to the options framework in hierarchical RL, except the ‘options’ here are linguistic subgoals. Results include pick-and-place block stacking experiments in the OpenAI Gym environment. Evaluations indicate that the proposed method outperforms naive RL agents that do not use subgoal decomposition.

Strengths:
+ Using language as a medium for reasoning about subgoals can be quite expressive. Language can be great for composing different parts of the sub-goal, in contrast to monolithic options that are less composable.
+ The experimental results include a good set of baselines including option-critic and planning-based approaches.

Weaknesses:
- The method explicitly supervises the language-based decomposition of goals into subgoals. In some sense, this extra supervision is a bit unfair with respect to naive RL agents or agents that have to ‘discover options’ from scratch.
- The methods section is missing implementation details: what is the action space? – is this joint control or end-effector control? How big is the observation space? Is it faster or slower to train with language-based sub-goals? – perhaps plots with steps vs. reward would be useful here. For the experiments, how many random seeds were these agents evaluated with, and what are the variances of the numbers in Table 2?
- The paper could also do a better job of highlighting the benefits of using language-based subgoals over using the options framework.  In some sense, the language sub-goals can also be considered as ‘options’. So what exactly is the difference between the Option-Critic and H Lang agents?
- The experiments could also be extended to other domains which involve more complex long-horizon tasks like in BabyAI or ALFRED different from just block-stacking.

Overall, the paper and topic is quite relevant to the workshop.

---

### Official Review · Reviewer_jJco · 2022-10-17
**Interesting problem statement, but experiments not convincing**

**Rating:** 6
**Confidence:** 4

**Review:**

Summary: The paper looks at solving long-horizon tasks, using hierarchical RL. The high-level policy takes a natural language goal and outputs subgoals represented using language. The low-level policy takes these subgoals to produce actions. Experiments include 3 versions of block manipulation -- 2 block rearrangement, 3 block rearrangement, and desk cleanup.

Strengths:
- The paper addresses an important problem -- generating a low-level plan from a high-level plan could be useful in many real-world applications.
- The paper is well-written and is easy to follow.

Weaknesses:
- The experiments are not very convincing, since the STRIPS planner outperforms the proposed approach. While the authors mention that the proposed approach is more interpretable and amenable to corrections, it is unclear that when the approach is applied to more realistic tasks, whether the drop in performance would be justified to get the interpretability and flexibility of corrections.
- For the tasks experimented with, it appears that the agent could complete them without access to language at all. For example, if the initial state has the blocks far apart, then they need to be moved closer, etc. It would be informative to see how the no-language baseline performs with the same architecture and training pipeline.

---

### Decision · Program_Chairs · 2022-10-21

Accept